# Surrogate-Based Robust Design for a Non-Smooth Radiation Source Detection Problem

**Răzvan Ştefănescu [1,2,\*]**, **Jason Hite [3]**, **Jared Cook [4]**, **Ralph C. Smith [4]** and **John Mattingly [3]**

[1] Global Validation Model Department, Spire Global, Inc., Boulder, CO 80301, USA
[2] Department of Computer Science, Virginia Tech, Blacksburg, VA 24060-0902, USA
[3] Department of Nuclear Engineering, North Carolina State University, Raleigh, NC 27695, USA; jmhite@ncsu.edu (J.H.); jkmattin@ncsu.edu (J.M.)
[4] Department of Mathematics, North Carolina State University, Raleigh, NC 27695, USA; jacook8@ncsu.edu (J.C.); rsmith@ncsu.edu (R.C.S.)
[\*] Correspondence: razstefanescu@gmail.com

**Abstract:** In this paper, we develop and numerically illustrate a robust sensor network design to optimally detect a radiation source in an urban environment. This problem exhibits several challenges: penalty functionals are non-smooth due to the presence of buildings, radiation transport models are often computationally expensive, sensor locations are not limited to a discrete number of points, and source intensity and location responses, based on a fixed number of sensors, are not unique. We consider a radiation source located in a prototypical 250 m × 180 m urban setting. To address the non-smooth properties of the model and computationally expensive simulation codes, we employ a verified surrogate model based on radial basis functions. Using this surrogate, we formulate and solve a robust design problem that is optimal in an average sense for detecting source location and intensity with minimized uncertainty.

**Keywords:** robust design in the average sense; Particle Swarm; radial basis functions; radiation source detection

## 1. Introduction

The problem of determining the location and intensity of a radiation source arises in several settings including emergency response to mitigate nuclear threats, structural and nuclear health monitoring in nuclear reactors, and environmental cleanup of biomedical and industrial nuclear waste. In this paper, we consider the development of a robust sensor network design for determining the location and intensity of a radiation source in a simulated urban environment. Specifically, we consider source localization in a simulated 250 m × 180 m block in downtown Washington, DC.

There are several difficulties that are intrinsic to this source localization problem. The first is that inverse problems of this nature are inherently ill-posed and require some form of regularization to obtain reasonable approximate solutions [1]. This difficulty is exacerbated by the fact that sensor observations are often coarsely spaced, which dictates that one cannot estimate source attributes that are more oscillatory than the grid spacing. As detailed in [2], this can yield erroneous results if ignored.

The computational complexity of deterministic [3] and stochastic, Monte Carlo [4] radiation transport models poses a second challenge since it limits the number of model realizations that can be obtained for optimization, or Bayesian or frequentist inference. This has led to the development of alternative parameterizations or surrogate models. For example, in [5] the authors modeled the radiation source as a point gamma source and employed a physics-based parameterization of gamma particle transport. A fast radiation transport model is also available as a component of

Synth, a gamma-ray simulation code written by Pacific Northwest National Laboratory [6,7]. In [8], the authors employ a Gaussian mixture to model the radiation field.

In [9], we addressed challenges associated with optimization and Bayesian inference as a prelude to the robust sensor design problem considered in this paper. Specifically, we implemented a fast piecewise-continuously differentiable radiation transport model and solved the associated inverse problem using combined global [10,11] and local [12] optimization algorithms, and Bayesian inference techniques [13,14]. As in [9], we assume here that the threat is a point source and that the model accounts only for photons that travel directly from source to detector, with no intervening collisions. The radiation source is parameterized with three components: its 2-D location coordinates and intensity.

To improve computational efficiency and permit gradient-based optimization and the implementation of a robust design algorithm [15], we implement and verify a continuously differentiable surrogate model based on radial basis functions to approximate the response for all possible detector locations. We employ this surrogate for subsequent optimization and robust design.

There are three different main strategies for taking measurements for general applications. The first searches for a given number of stationary sensors, the second one relies on moving sensors, whereas the third method, entitled scanning, activates only a subset of a given number of stationary sensors at a given moment in time. The existing methods for identification of sensor locations usually employ random fields analysis [16], information theory [17] and optimum experimental design theory [15]. Moreover, sensor placement algorithms can be classified into discrete and continuous depending on the nature of the search space. In the context of nuclear source identification, Michaud [18] used the Gaussian process optimization [19] to solve a continuous detector placement problem and Schmidt [20] applied Shannon entropy [21–23] to guide mobile sensors over a discrete grid of possible measurement sites.

To form a basis for comparing different networks, a quantitative measure of efficiency is required. In this study, we explore criteria applied in optimum experimental design [15] to solve a discrete stationary detector placement problem. These criteria are defined in terms of the Fisher information matrix associated with the unknown characteristics of the source. One of the main difficulties associated with optimization of sensor locations is the dependence of the optimal solutions on unknown true values of the source characteristics or prior approximations. To remove this dependency, we employ a robust design strategy based on maximizing the expectation of the corresponding local optimality criterion over the source characteristics domain. We then transform the resulting stochastic optimization problem into a combinatorial optimization problem by generating a finite set of possible detectors locations. We solve the combinatorial optimization problem and test the obtained optimal network of sensors against randomly selected networks. The surrogate model implementation allows solving the combinatorial optimization problem otherwise being computationally infeasible.

The remainder of the paper is organized as follows. In Section 2, we discuss the radiation transport model and radial basis function surrogate along with associated statistical models. We also describe the domain geometry. The inverse problem based on the surrogate models is formulated in Section 3. In Section 4, we present the theoretical framework of the robust design in the average sense employed in this paper. In Section 5, we present the numerical solution of the robust design problem and compare the optimal network performance to randomly selected networks. We draw conclusions in Section 6. We summarize in the Appendix A the Particle Swarm algorithm used to solve the inverse problem.

## 2. Radiation Transport Model and Surrogate Formulations

Gamma transport phenomena, as derived from Boltzmann transport theory, can be modeled by the partial differential equation (PDE)

$$
\hat{\mathbf{\Omega}} \cdot \nabla I(\mathbf{r}, E, \hat{\mathbf{\Omega}}) + \Sigma_t(\mathbf{r}, E) I(\mathbf{r}, E, \hat{\mathbf{\Omega}}) \\
= S(\mathbf{r}, E) + \int_0^\infty dE' \int_{4\pi} d\hat{\mathbf{\Omega}}' \Sigma_s(\mathbf{r}, E' \to E, \hat{\mathbf{\Omega}}' \to \hat{\mathbf{\Omega}}) I(\mathbf{r}, E', \hat{\mathbf{\Omega}}').
$$

(1)

Here $I$ and $S$ respectively denote the gamma intensity per unit area and external gamma source in the medium characterized by the position vector $\mathbf{r}$, energy $E$, and unit vector in the direction of the gamma $\hat{\Omega}$. The parameters include the total macroscopic cross-section for gamma interactions $\Sigma_t$, and the double-differential macroscopic scattering cross-section $\Sigma_s$, which depends on the change in gamma energy from incident energy $E'$ to emergent energy $E$ (i.e., $E' \rightarrow E$) and the change in gamma direction from incident direction $\Omega'$ to $\Omega$ (i.e., $\Omega' \rightarrow \Omega$). We refer readers to Shultis and Faw [3] for a more detailed treatment of transport theory.

The problem of inferring the radiation source location and intensity from sensor measurements requires the evaluation of the Boltzmann radiation transport model (1) at various points in the admissible parameter space. Numerically solving the PDE (1) is computationally expensive even on HPC systems. Solving an inverse problem constrained by Equation (1) or forward propagating uncertainties using Monte Carlo simulations are not computationally feasible since require solving Equation (1) for many times.

Instead, we employ a model that only considers gamma rays that travel directly from source to detectors, without taking into account photons that incur collisions. This approach relies on the assumption that photons undergoing interactions in the medium have a very small probability of ever arriving at a detector. We also assume that the physical scale of our problem is sufficiently large so that both the source and detectors can be localized to points inside the domain. We will denote the location of the source as $\mathbf{r}_s$ and associated intensity by $S_0$. $S_0$ can be treated as time-independent. Most radionuclides of interest for source search have half-lives on the order of several years to tens-of-thousands of years. Consequently, radioactive decay of the source is insignificant during the measurement. Under these assumptions, Equation (1) can be simplified to

$$\hat{\Omega} \cdot \nabla I(\mathbf{r}, E, \Omega) + \Sigma_t(\mathbf{r}, E, \hat{\Omega}) I(\mathbf{r}, E, \hat{\Omega}) = \frac{S_0}{4\pi} \delta(E - E_0) \delta(\|\mathbf{r} - \mathbf{r}_s\|_2), \tag{2}$$

where $\hat{\Omega}$ is a unit vector pointing in the traveling direction of the gamma rays and $E_0$ is the source emission energy and delta denotes the Dirac delta function; see [3] for more details. Equation (2) can be solved to determine the intensity of photons arriving at any point $\mathbf{r}$ inside domain. This enables the computation of the count rate measured by the $i$-th detector $D_i$ assuming that detectors are point detectors with face area $A_i$ and dwell time $\Delta t_i$. The detector intrinsic efficiency $\epsilon_i \in [0, 1]$ is usually known in practice.

If the *ith* detector is located at point $\mathbf{r}_d^i$, the solution

$$\hat{F}_i : \mathcal{X} \to \mathbb{R}, \hat{F}_i(\boldsymbol{\theta}) = S_0 \Delta t_i \cdot \epsilon_i \cdot \frac{A_i}{4\pi \|\mathbf{r}_d^i - \mathbf{r}_s\|_2^2} \cdot \exp\left(\int_{\mathbf{r}_d^i - \mathbf{r}_s} \Sigma_T \, d\mathbf{s}\right) \tag{3}$$

of Equation (2) predicts the number of counts observed by the sensor given the location and intensity $\boldsymbol{\theta} = (\mathbf{r}_s, S_0)$ of the source. Here we denoted by $\mathcal{X}$ the space of all possible sources and $\mathbb{R}$ is the one-dimensional real coordinate space. The derivation of model response (3) follows in a manner similar to that shown in Shultis and Faw [3], (Chapter 10.1.3), where the resulting solution is evaluated at the detector location $\mathbf{r}_d^i$.

## 2.1. Model Geometry

To provide an example of an urban area, we selected a 250 m × 180 m block in downtown Washington, D.C., located at approximately 38°54′48″ N by 77°1′60″ W (Johnson Avenue NW) to serve as our domain. Buildings in this area are primarily brick and concrete residential housing and are generally 1–5 stories in height. Using data from the OpenStreetMaps database (https://www.openstreetmap.org/), we constructed a 2-D representation of the area to serve as the test geometry. Our implementation treats the buildings as a set of disjoint polygons $P_j$, $j = 1, 2, ..., N_g$, each of which

is assigned a corresponding macroscopic cross-section $\Sigma_t$. A satellite photo of the area with an overlay of the constructed representation is provided in Figure 1.

Approximate calculations indicate that wood and concrete buildings correspond to an optical thickness of around 3 mean free paths (MFPs), where the mean free path denotes the mean distance traveled by the photons between collisions with atoms of the building. Consequently, we randomly selected cross-sections for each building so that their optical thickness is between 1 and 5 MFPs. The random sampling was also weighted according to the volume of each building, so that smaller buildings were biased towards smaller optical thicknesses and vice versa. The regions between buildings were treated as dry air at standard temperature and pressure, with cross-sections taken from the NIST XCOM database (http://www.nist.gov/pml/data/xcom/).

For this geometry, the admissible parameter space is

$$\mathcal{X} = [0, 250] \times [0, 180] \times [5 \cdot 10^8, 5 \cdot 10^{10}]. \tag{4}$$

The first two dimensions define the spatial location representing the simulated $250 \times 180$ m urban block. The third dimension restricts the source intensity to vary between $5 \times 10^8$ and $5 \times 10^{10}$ Bq.

### 2.2. Numerical Model for Detector Response

To determine the intensity of photons arriving at a given detector location $\mathbf{r}_d^i$, the algorithm employs a simple ray-tracing scheme. Starting at the location of the source $\mathbf{r}_s$, we draw a ray from $\mathbf{r}_s$ to $\mathbf{r}_d^i$. We then compute the intersection of this ray with the disjoint polygons $P_j$, $j = 1, 2, ..., N_g$, representing the set of buildings in our domain. This yields a series of line segments expressing the path traversed in each region. We assume that a given ray intersects $N_\ell$ polygons, $N_\ell < N_g$, and let $\mathcal{L} = \{(\ell_j, \Sigma_T^{(j)})\}_{j=1}^{N_\ell}$ be the set of all intersecting segments, where $\ell_j$ is the Euclidean length of the $j$-th segment and $\Sigma_T^{(j)}$ is the corresponding value for the macroscopic total cross-section. With this assumption, Equation (3) takes the form

$$\hat{F}_i(\boldsymbol{\theta}) = S_0 \Delta t_i \cdot \epsilon_i \cdot \frac{A_i}{4\pi \|\mathbf{r}_d^i - \mathbf{r}_s\|_2^2} \exp\left( -\sum_{j=1}^{N_\ell} \ell_j \cdot \Sigma_T^{(j)} \right). \tag{5}$$

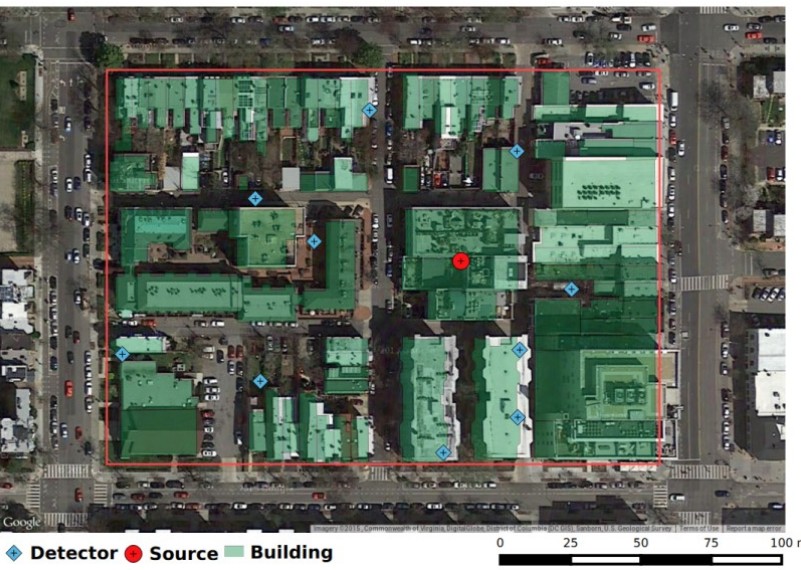

◆ **Detector** ● **Source** ▇ **Building**

**Figure 1.** Satellite image of problem domain with model geometry overlaid (Imagery ©2016 Commonwealth of Virginia, DigitalGlobe, District of Columbia (DC GIS), Sanborn, U.S. Geological Survey, Map data ©2016 Google). The figure marks indicate possible detector and source locations.

Equation (5) provides an analytic expression estimating the expected detector response, and its computation primarily requires the intersection of lines with the model geometry. Equation (5) represents a significant simplification to the solution of (1), a nonlinear PDE with seven independent variables whose solution in complex geometries can require many hours even on a supercomputer. We implemented the numerical model (5) in a short Python code. It employs the Shapely library (https://pypi.python.org/pypi/Shapely) for performing the computational geometry calculations. The model takes as input a specification of polygons representing the different regions of the domain, cross-section data, detector locations, source intensity, and source location.

### 2.3. Statistical Model

To construct statistical models associated with $N$ detectors, we consider a background with constant expected intensity $B$. We denote by $\boldsymbol{\theta}_0$, the true source location and intensity of a radiation source. It is well known that radioactive decay and detection are Poisson random processes. By including Poisson random effects and assuming that $N$ detectors are available, we obtain the statistical model

$$Y_i \sim \mathrm{P}\left(\hat{F}_i(\boldsymbol{\theta}_0) + B\right), \tag{6}$$

associated with the *ith* detector response, $i = 1, \ldots, N$. The Poisson distribution with mean

$$F_i : \mathcal{X} \to \mathbb{R}, \; F_i(\boldsymbol{\theta}_0) = \hat{F}(\boldsymbol{\theta}_0) + B \tag{7}$$

is denoted by $\mathrm{P}$. For large numbers (>30) of observed photons, the Poisson distribution is adequately approximated by a normal distribution, yielding the approximate statistical model

$$Y_i \sim \mathcal{N}\left(F_i(\boldsymbol{\theta}_0), (\sigma_i^o)^2\right), \tag{8}$$

where $(\sigma_i^o)^2 = F_i(\boldsymbol{\theta}_0)$; i.e., with variance equal to the mean. This is equivalent to

$$Y_i = F_i(\boldsymbol{\theta}_0) + \varepsilon_i^o, \; \varepsilon_i^0 \sim \mathcal{N}(0, (\sigma_i^o)^2). \tag{9}$$

In this manner, we model the observations associated with each detector as random variables $Y_i$, $i = 1, \ldots, N$.

### 2.4. Radial Basis Function Surrogate Model

Due to the presence of the buildings, the model response (7) is non-differentiable with respect to both position and intensity. To apply sensitivity analysis to determine an optimal sensor configuration, smoothness of the model responses must be assured. To address these issues and reduce computational times, we used radial basis functions to provide continuously differentiable approximations of the model responses (7).

Radial basis function methods provide interpolants to sampled values associated with irregularly positioned points inside the input domain. A radial basis function approximation of the model response $F_i(\boldsymbol{\theta})$ has the formulation

$$\tilde{F}_i : \mathcal{X} \to \mathbb{R}, \tilde{F}_i(\boldsymbol{\theta}) = \sum_{k=1}^{\mathcal{L}} \lambda_k \psi(\varepsilon \|\boldsymbol{\theta} - \boldsymbol{\theta}_k\|_2), \tag{10}$$

where $\boldsymbol{\theta}$ denotes a source in the domain $\mathcal{X}$, $\psi : \mathbb{R} \to \mathbb{R}$ is a radial basis function and $\varepsilon$ is a shape parameter. Possible choices of radial basis functions $\psi$ include multiquadrics and their inverse formulations, Gaussian functions, and thin plate splines. A more comprehensive list can be found in [14,24]. We employ Gaussian radial basis functions. The coefficients $\lambda_k$ are computed by requiring that $\tilde{F}_i(\boldsymbol{\theta}_k) = F_i(\boldsymbol{\theta}_k), k = 1, \ldots, \mathcal{L}$, where $\boldsymbol{\theta}_k$ are selected to cover the entire domain $\mathcal{X}$ and $\mathcal{L}$ is the number of interpolation points. We employed the MATLAB radial basis function toolbox based on

Cholesky factorization and Tikhonov regularization. We also tried other methods to approximate the model response based on Legendre and Lagrange polynomials and Gaussian process. Our results (not shown here) revealed that the radial basis functions approximation had the best accuracy for our application.

*2.5. Surrogate Statistical Model*

The analysis of interpolation error relies on smoothness properties of the map being approximated. In our case, such properties are not directly available nor are error bounds. Instead, we assume that the response surrogate models errors associated with the true source can be modeled as normal random variables $\varepsilon_i^m \sim \mathcal{N}(0, (\sigma_i^m)^2)$, yielding the statistical model

$$F_i(\boldsymbol{\theta}_0) = \tilde{F}_i(\boldsymbol{\theta}_0) + \varepsilon_i^m, \ i = 1, \ldots, N, \tag{11}$$

with $\sigma_i^m$ being the standard deviation of the response surrogate models errors.

A statistical model incorporating both model errors $\varepsilon_i^m$ and observation errors $\varepsilon_i^o$ introduced in (9), is

$$Y_i = \tilde{F}_i(\boldsymbol{\theta}_0) + \varepsilon_i^o + \varepsilon_i^m, \ i = 1, \ldots, N. \tag{12}$$

Assuming the independence of model errors and observation errors, and exploiting the fact that the sum of independent normal random variables is also a normal random variable, (12) can be expressed as

$$Y_i = \tilde{F}_i(\boldsymbol{\theta}_0) + \tilde{\varepsilon}_i, \ Y_i \sim \mathcal{N}(\tilde{F}_i(\boldsymbol{\theta}_0), \sigma_i^2) \ i = 1, \ldots, N, \tag{13}$$

where $\tilde{\varepsilon}_i \sim \mathcal{N}(0, \sigma_i^2)$, and $\sigma_i^2 = (\sigma_i^o)^2 + (\sigma_i^m)^2$.

## 3. Detection of Nuclear Radiation Sources Using Surrogate Model

The problem of estimating the location and intensity of a radiation source when several detectors with associated measurements are available represents a classic inverse problem. The source $\boldsymbol{\theta}_0$ is unknown and has to be inferred from realizations $v_i$ of the random variables $Y_i$, $i = 1, \ldots N$ whose statistical model is described in (13).

The Gaussian likelihood function $\pi : \mathcal{X} \to [0, \infty)$ is given by

$$\pi(\boldsymbol{V}|\boldsymbol{\theta}) = \frac{1}{\sqrt{(2\pi)^N \prod_{i=1}^N \sigma_i}} \exp\left( - \sum_{i=1}^N \frac{1}{\sigma_i^2} [v_i - \tilde{F}_i(\boldsymbol{\theta})]^2 \right), \tag{14}$$

where $\boldsymbol{V} = [v_1, \ldots, v_N]$ is the vector of all the available observations.

A standard technique to estimate the location and intensity of a radiation source, based on measured data, is to apply maximum likelihood estimators. Due to the monotonicity of the logarithm function, maximizing (14) is equivalent to minimizing the negative logarithm of the likelihood

$$\min_{\boldsymbol{\theta} \in \mathcal{X}} J(\boldsymbol{\theta}), \ J(\boldsymbol{\theta}) = \frac{1}{2} \sum_{i=1}^N \frac{1}{\sigma_i^2} [v_i - \tilde{F}_i(\boldsymbol{\theta})]^2. \tag{15}$$

We omit the constant in front of the exponential term in Equation (14) since it does not affect the solution. The objective of this investigation is to solve the optimization problem (15) with minimal uncertainty. The solution to this problem (15) is a maximum likelihood estimator which is a random variable. In this context, uncertainty represents the covariance of this estimator which is shown to be the inverse of the Fisher information matrix (23). To achieve this, we apply the robust design strategy described in next section.

## 4. Robust Design in the Average Sense

Here we present a robust design strategy in the average sense. The solution of the robust design problem is a network of sensors that minimizes the uncertainty in the solution of the inverse problem (15).

### 4.1. Solution of the Inverse Problem

The solution of a nonlinear inverse problem is typically more difficult than in the linear case since one regularly does not have analytic solutions. Under the assumption that the model (10) behaves linearly with respect to variables $\boldsymbol{\theta}$, one can derive an analytic solution for the weighted least-square estimate as described in [15,25]. For nonlinear model responses, it is customary to linearize the system response

$$\tilde{F}_i(\boldsymbol{\theta}) = \tilde{F}_i(\boldsymbol{\theta}^0) + \frac{\partial \tilde{F}_i}{\partial \boldsymbol{\theta}}(\boldsymbol{\theta}^0)^T(\boldsymbol{\theta} - \boldsymbol{\theta}^0) + \mathcal{O}(\|\boldsymbol{\theta} - \boldsymbol{\theta}^0\|_2^2), \; i = 1, \dots, N \tag{16}$$

about a prior estimate $\boldsymbol{\theta}^0$, where $\| \cdot \|_2$ is the Euclidean norm.

By neglecting the higher order terms, the statistical model (13) can be rewritten as

$$\tilde{Y}_i = \frac{\partial \tilde{F}_i}{\partial \boldsymbol{\theta}}(\boldsymbol{\theta}^0)^T\boldsymbol{\theta}_0 + \tilde{\varepsilon}_i, \; i = 1, \dots, N. \tag{17}$$

Here a realization $\tilde{v}_i$ of $\tilde{Y}_i$ can be expressed as $\tilde{v}_i = v_i - \tilde{F}_i(\boldsymbol{\theta}^0) + \frac{\partial \tilde{F}_i}{\partial \boldsymbol{\theta}}(\boldsymbol{\theta}^0)^T\boldsymbol{\theta}^0$, where $v_i$ is sampled from the distribution defined in (13). Consequently, the problem (15) can be reformulated as

$$\hat{\boldsymbol{\theta}} = \arg \min_{\boldsymbol{\theta}} \mathcal{J}(\boldsymbol{\theta}), \; \mathcal{J}(\boldsymbol{\theta}) = \frac{1}{2}\sum_{i=1}^{N}\frac{1}{\sigma_i^2}\left[\tilde{v}_i - \frac{\partial \tilde{F}_i}{\partial \boldsymbol{\theta}}(\boldsymbol{\theta}^0)^T\boldsymbol{\theta}\right]^2. \tag{18}$$

By differentiating $J$ with respect to $\boldsymbol{\theta}$, we obtain

$$\nabla J(\hat{\boldsymbol{\theta}}) = -\sum_{i=1}^{N}\frac{1}{\sigma_i^2}\frac{\partial \tilde{F}_i}{\partial \boldsymbol{\theta}}(\boldsymbol{\theta}^0)\left[\tilde{v}_i - \frac{\partial \tilde{F}_i}{\partial \boldsymbol{\theta}}(\hat{\boldsymbol{\theta}}^0)^T\boldsymbol{\theta}\right]. \tag{19}$$

By imposing $\nabla J(\hat{\boldsymbol{\theta}}) = 0$, the estimate

$$\hat{\boldsymbol{\theta}} = \boldsymbol{M}^{-1}\sum_{i=1}^{N}\frac{1}{\sigma_i^2}\frac{\partial \tilde{F}_i}{\partial \boldsymbol{\theta}}(\boldsymbol{\theta}^0)\tilde{v}_i, \tag{20}$$

is unique only if the Fisher information matrix

$$\boldsymbol{M} = \sum_{i=1}^{N}\frac{1}{\sigma_i^2}\frac{\partial \tilde{F}_i}{\partial \boldsymbol{\theta}}(\boldsymbol{\theta}^0)\left[\frac{\partial \tilde{F}_i}{\partial \boldsymbol{\theta}}(\boldsymbol{\theta}^0)\right]^T \tag{21}$$

is nonsingular.

A formula for the estimator $\boldsymbol{\theta}$ of the problem (15) is obtained based on the linearization of the surrogate model response (10) and by assuming that the high-order terms in the expansion (16) are negligible. In the next subsection, we highlight the role of Fisher information matrix in defining the D-optimality criterion widely used in the optimum experimental design problems as a quantitative measure of the 'goodness' of different networks of sensors.

### 4.2. Optimal Sensor Locations

We focus on the statistical properties of the maximum likelihood estimator whose estimate was derived in Equation (20). Based on these properties, optimal design theory provides solutions for the optimal placement of sensors problem.

We first note that the estimator

$$\tilde{\boldsymbol{\theta}} = \boldsymbol{M}^{-1} \sum_{i=1}^{N} \frac{1}{\sigma_i^2} \frac{\partial \tilde{F}_i}{\partial \boldsymbol{\theta}}(\boldsymbol{\theta}^0)\tilde{Y}_i, \tag{22}$$

is unbiased since

$$E[\tilde{\boldsymbol{\theta}}] = \boldsymbol{M}^{-1} \sum_{i=1}^{N} \frac{1}{\sigma_i^2} \frac{\partial \tilde{F}_i}{\partial \boldsymbol{\theta}}(\boldsymbol{\theta}^0) E[\tilde{Y}_i] = \boldsymbol{M}^{-1} \sum_{i=1}^{N} \frac{1}{\sigma_i^2} \frac{\partial \tilde{F}_i}{\partial \boldsymbol{\theta}}(\boldsymbol{\theta}^0) E\left[ \frac{\partial \tilde{F}_i}{\partial \boldsymbol{\theta}}(\boldsymbol{\theta}^0)^T \boldsymbol{\theta}_0 + \tilde{\varepsilon}_i \right],$$

$$= \boldsymbol{M}^{-1} \sum_{i=1}^{N} \frac{1}{\sigma_i^2} \frac{\partial \tilde{F}_i}{\partial \boldsymbol{\theta}}(\boldsymbol{\theta}^0) \frac{\partial \tilde{F}_i}{\partial \boldsymbol{\theta}}(\boldsymbol{\theta}^0)\boldsymbol{\theta}_0 = \boldsymbol{\theta}_0,$$

and $\boldsymbol{\theta}_0$ is the true source location and intensity of the radiation source. Please note that we employed the relation $E[\tilde{\varepsilon}_i] = 0$, $i = 1, \dots N$, and Formula (21).

The covariance of the estimator is given by

$$\begin{aligned}
\text{cov}[\tilde{\boldsymbol{\theta}}] &= E[(\tilde{\boldsymbol{\theta}} - \boldsymbol{\theta}_0)(\tilde{\boldsymbol{\theta}} - \boldsymbol{\theta}_0)^T] \\
&= E\left[ (\boldsymbol{M}^{-1} \sum_{i=1}^{N} \frac{1}{\sigma_i^2} \frac{\partial \tilde{F}_i}{\partial \boldsymbol{\theta}}(\boldsymbol{\theta}^0)\tilde{Y}_i - \boldsymbol{\theta}_0)(\boldsymbol{M}^{-1} \sum_{j=1}^{N} \frac{1}{\sigma_i^2} \frac{\partial \tilde{F}_j}{\partial \boldsymbol{\theta}}(\boldsymbol{\theta}^0)\tilde{Y}_j - \boldsymbol{\theta}_0)^T \right] \\
&= \boldsymbol{M}^{-1} \sum_{i=1}^{N} \sum_{j=1}^{N} \frac{1}{\sigma_i^2} \frac{\partial \tilde{F}_i}{\partial \boldsymbol{\theta}}(\boldsymbol{\theta}^0) E\left[ \tilde{\varepsilon}_i \tilde{\varepsilon}_j \right] \frac{1}{\sigma_j^2} \frac{\partial \boldsymbol{F}_j}{\partial \boldsymbol{\theta}}(\boldsymbol{\theta}^0)^T \boldsymbol{M}^{-T} \\
&= \boldsymbol{M}^{-1} \sum_{i=1}^{N} \frac{1}{\sigma_i^2} \frac{\partial \tilde{F}_i}{\partial \boldsymbol{\theta}}(\boldsymbol{\theta}^0) E\left[ \tilde{\varepsilon}_i^2 \right] \frac{1}{\sigma_i^2} \frac{\partial \tilde{F}_i}{\partial \boldsymbol{\theta}}(\boldsymbol{\theta}^0)^T \boldsymbol{M}^{-T} = \boldsymbol{M}^{-1}.
\end{aligned} \tag{23}$$

The result is based on the independence of the errors and symmetry of $\boldsymbol{M}$.

The Fisher information matrix does not depend on the pseudo-measurements $\bar{v}_i$ but instead on the sensor locations $\boldsymbol{D}_i$, $i = 1, \dots, N$, and the prior estimate $\boldsymbol{\theta}^0$. As such, one can adjusts the sensor locations $\boldsymbol{D}_i$ to minimize uncertainty in the estimator; i.e., minimize $\text{cov}[\tilde{\boldsymbol{\theta}}]$. As detailed in [26], an optimal configuration exists only for specific cases. This is the reason for the introduction of scalar metrics depending on all possible Fisher information matrices [26]. The most popular metrics are the D-optimality, E-optimality, A-optimality, and sensitivity criteria based on the determinant, smallest eigenvalue, and trace of the Fisher information matrix and its inverse. D-optimality is invariant under scale changes in the parameters and linear transformations of the output in contrast to A-optimality and E-optimality criteria.

In this investigation, we employ the D-optimality criterion, where the optimal network $\boldsymbol{\xi}_N^* = \{\boldsymbol{D}_i^*, i = 1, \dots, N\}$ is searched as the solution of the optimization problem

$$\max_{\boldsymbol{\xi}_N} \Psi\left( \boldsymbol{M}(\boldsymbol{\xi}_N, \boldsymbol{\theta}^0) \right), \Psi\left( \boldsymbol{M}(\boldsymbol{\xi}_N, \boldsymbol{\theta}^0) \right) = \det\left( \boldsymbol{M}(\boldsymbol{\xi}_N, \boldsymbol{\theta}^0) \right). \tag{24}$$

Here $\boldsymbol{\xi}_N$ ranging in the space of all possible combinations of sensor locations.

### 4.3. Robust Design in the Average Sense

The optimal design $\boldsymbol{\xi}_N^*$ obtained as the solution of (24) is dependent on the prior estimate $\boldsymbol{\theta}^0$ of the true parameters $\boldsymbol{\theta}_0$. When the prior estimate $\boldsymbol{\theta}^0$ is not a reasonable approximation of $\boldsymbol{\theta}_0$, the network $\boldsymbol{\xi}_N^*$ may be inaccurate since prior uncertainty on $\boldsymbol{\theta}^0$ is not taken into account. For the admissible parameter space $\mathcal{X}$ in (4), the set of all possible characteristics $\boldsymbol{\theta}$, is compact. By incorporating the probabilistic description of the prior uncertainty, we obtain the optimal design in the average sense. The quantity of interest to be maximized is the expectation of the corresponding local optimality criterion,

$$\Gamma(\boldsymbol{\xi}_N) = E_{\boldsymbol{\theta}}\left[\Psi[\boldsymbol{M}(\boldsymbol{\xi}_N,\boldsymbol{\theta})]\right] = \int_{\mathcal{X}} \Psi[\boldsymbol{M}(\boldsymbol{\xi}_N,\boldsymbol{\theta})]p(\boldsymbol{\theta})d\boldsymbol{\theta}, \tag{25}$$

where $p(\boldsymbol{\theta})$ is the uniform distribution on $\mathcal{X}$. We employ the criterion introduced in (24) leading to the ED-optimal design problem

$$\max_{\boldsymbol{\xi}_N} \Gamma_{ED}, \Gamma_{ED}(\boldsymbol{\xi}_N) = \int_{\mathcal{X}} \det\left(\boldsymbol{M}(\boldsymbol{\xi}_N,\boldsymbol{\theta})\right)p(\boldsymbol{\theta})d\boldsymbol{\theta}. \tag{26}$$

## 5. Numerical Examples

As detailed in Section 3, the problem under investigation consists of identifying the location and intensity of a radiation source in a simulated downtown Washington, DC block with minimum error with respect to the true source location and intensity. The solution to this problem can be obtained by applying optimal sensor location strategies [15]. Instead of applying a local optimal design method, whose solution depends on some *a priori* estimate of the true source, we propose a robust design strategy to remove this dependency. Specifically, we propose a 'compromise' design where the obtained network is good enough (in a least-error sense) to identify any possible source from the admissible domain $\mathcal{X}$.

As detailed in (4), we take the admissible parameter space to be $\mathcal{X} = [0,250]$ m $\times$ $[0,180]$ m $\times$ $[5 \cdot 10^8, 5 \cdot 10^{10}]$ Bq. Next we specify the set of all possible detector locations to be a discrete set of 30 spatial positions. By sampling from a uniform distribution, we generate the possible locations of detectors in the domain denoted by diamond marks in Figure 2. In this way, we avoid the problem of overlapping sensors encountered for a continuous formulation. The specific dispersal pattern was selected to spread the detectors evenly throughout the area. We assume that detectors have facial areas $A_i$, with 3-inch diameters and 3-inch lengths, for incident gamma energy of 662 KeV. This is standard packaging for sodium iodide (NaI) scintillators that possess intrinsic efficiency of $\varepsilon_i = 62\%$ for 662 keV gammas. The dwell time $\Delta t_i$ for all detectors was chosen to be 1 s.

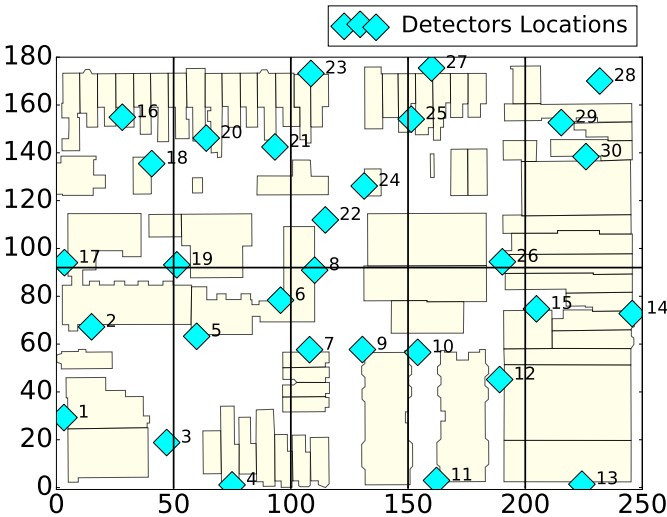

**Figure 2.** Network formed by 10 sensors with 30 possible locations.

Finally, we set the size of the network to 10 detectors and formulate the robust design problem in the average sense:

*Find the network* $\boldsymbol{\xi}_{10}^* = \{\boldsymbol{D}_i, i = 1, \ldots, 10\}$ *consisting of* 10 *detectors out of the* 30 *possible detectors locations depicted in Figure* 2 *that solves*

$$\boldsymbol{\xi}_{10}^* = argmax_{\boldsymbol{\xi}_{10}}\Gamma_{ED}, \quad \Gamma_{ED}(\boldsymbol{\xi}_{10}) = \int_{\mathcal{X}} \det\left(\boldsymbol{M}(\boldsymbol{\xi}_{10},\boldsymbol{\theta})\right)p(\boldsymbol{\theta})d\boldsymbol{\theta}. \tag{27}$$

By using a sufficiently large number of sources, the integral in (27) can be accurately approximated. To evaluate the efficiency of the robust design network, we employ the metric

$$
\sqrt{\frac{\sum_{\ell=1}^{\mathcal{M}} \|\hat{\boldsymbol{\theta}}_{\boldsymbol{\xi}_{10}}^{\ell} - \boldsymbol{\theta}_0^{\ell}\|_2^2}{\mathcal{M}}}, \tag{28}
$$

where $\boldsymbol{\theta}_0^{\ell}$, $\ell = 1, \cdots \mathcal{M}$, are $\mathcal{M}$ distinct true radiation sources. For each true source $\boldsymbol{\theta}_0^{\ell}$, we can compute the associated estimate $\hat{\boldsymbol{\theta}}_{\boldsymbol{\xi}_{10}}$ as the solution of the problem (15) using the network $\boldsymbol{\xi}_{10}$. The score (28) corresponding to the robust design $\boldsymbol{\xi}_{10}^*$ will be tested against scores obtained by randomly selected networks $\boldsymbol{\xi}_{10}$ of 10 detectors. The smallest score should be obtained by the robust design $\boldsymbol{\xi}_{10}^*$ thus validating the approach.

The discrete nature of the space of all possible detectors locations transforms (27) into a combinatorial optimization problem. The number of possible networks is 30,045,015 as given by (30 choose 10) which is equivalent to combination of 30 possible sensor locations taken 10. By imposing that each possible network $\boldsymbol{\xi}_{10}$ contains only one detector out of the three possible choices from each of the ten rectangular areas shown in Figure 2, we decrease the number of possible networks to 59,049. This makes the combinatorial problem computationally feasible.

In Figure 3, we plot the model response $F_8$ (7) corresponding to the sensor location $\boldsymbol{D}_8$ in Figure 2 and a source intensity of $3.5 \times 10^9$ Bq. The source location is varied inside the domain and the non-smooth nature of the model response is observed. The Fisher information matrix requires that the model response be differentiable with respect to the source location and intensity. This motivates replacing the model responses for all 30 possible locations with the differentiable radial basis function surrogate model described in Section 2.4.

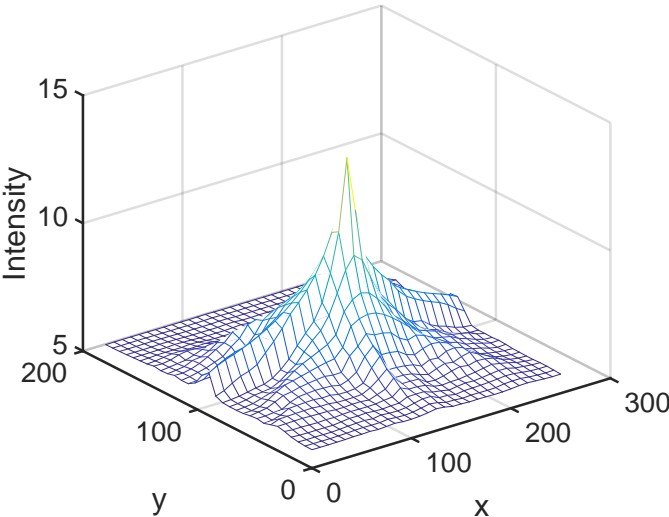

**Figure 3.** Model response associated with sensor location number 8 with the source intensity fixed at $3.5 \times 10^9$ Bq. Intensity is plotted with a logarithmic scale.

We employ radial basis interpolation (10) to generate 30 surrogate model responses for all the possible detectors locations. The number of interpolation points is selected at 29,791 distributed inside the domain $\mathcal{X}$. Specifically, for each dimension, we selected 31 points evenly distributed inside the interval. For each possible detector location and source $\boldsymbol{\theta}_k$, the response model $F_i$ (7) was used to calculate the corresponding interpolation points $(\boldsymbol{\theta}_k, F_i(\boldsymbol{\theta}_k))$, $k = 1, \ldots, 29{,}791$, $i = 1, \ldots, 30$. We tested several values of the shape parameter $\varepsilon$ and the most accurate surrogate models were obtained for $\varepsilon^p = 8.06$.

In Figure 4a, we compare the surrogate model predictions against the outputs of model response $F_8$ for $\boldsymbol{D}_8$ and 100 different sources uniformly randomly sampled from $\mathcal{X}$. These sources were not

included in the training set. The red curve corresponds to the model response outputs whereas the blue curve denotes the surrogate model predictions. Figure 4b illustrates the relative errors of the predicted intensities.

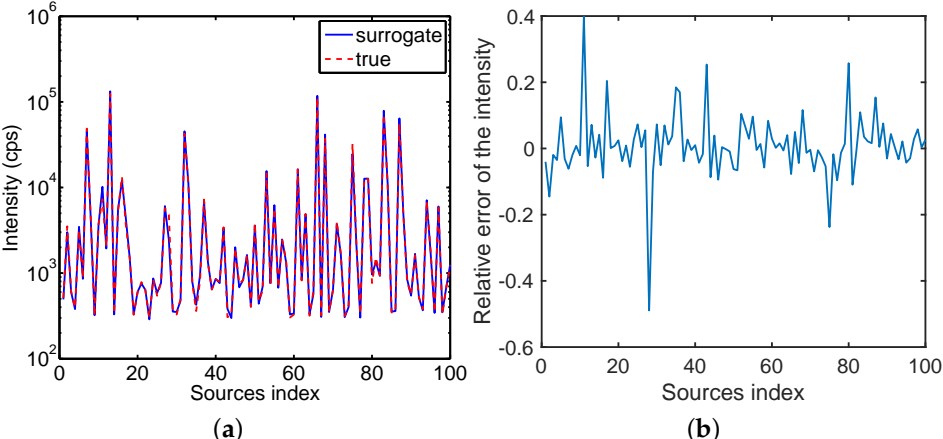

|     |     |
| :-: | :-: |
| (a) | (b) |

**Figure 4.** (**a**) Predictions of the surrogate and model responses associated with sensor location number 8. (**b**) Relative errors of the predicted intensity. The test set consists of 100 different sources not included in the training set.

To compute accurate variances for each of the 30 surrogate model responses, we generated a data set of $10^5$ different sources uniform randomly spread inside domain $\mathcal{X}$. The root mean square errors (RMSE) are shown in Figure 5 for all 30 surrogate models. The largest error is observed for the surrogate model response associated with $D_3$. We note that whereas the source intensity ranges between $5 \times 10^8$ and $5 \times 10^{10}$ Bq, the largest RMSE is on the order of $4.2 \times 10^6$ counts per second (cps). The discrepancies between the outputs of the models $\tilde{F}_i$ and $F_i$ are then used to compute variances $(\sigma_i^m)^2$, $i = 1, \ldots, 30$.

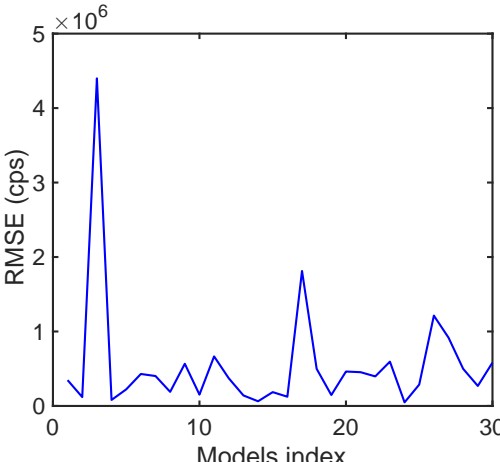

**Figure 5.** Root mean square error of all 30 surrogate models. The test set consists of $10^5$ different sources not included in the training set.

Next we generated observations for all possible networks $\boldsymbol{\xi}_{10}^{\ell}, \ell = 1, \ldots, 59{,}049$, using statistical model (9) based on the model response discussed in Section 2.2. The observation errors associated with each detector and source $\boldsymbol{\theta}$ are normally distributed with mean 0 and variance $(\sigma_i^o)^2 = F_i(\boldsymbol{\theta})$, $i = 1, \ldots, 30$.

The robust design problem solution is given by the network $\boldsymbol{\xi}_{10}$ associated with the largest score $\Gamma_{ED}$ (27). To determine its maximum value, we calculate the associated Fisher information matrix

for all possible networks $\boldsymbol{\xi}_{10}^{\ell}$, $\ell = 1, \ldots$, 59,049 and a collection of possible sources $\boldsymbol{\theta}_{\ell}, \ell = 1, \ldots, 9880$ spread throughout the domain $\mathcal{X}$. This allows us to estimate the integral in (27). The dependencies associated with the Fisher information matrix are the derivatives of the surrogate models with respect to the source characteristics and variances $(\sigma_i)^2 = (\sigma_i^o)^2 + (\sigma_i^m)^2$, $i = 1, \ldots, 30$. The gradients $\frac{\partial \tilde{F}_i}{\partial \boldsymbol{\theta}_l}$ are computed from (10) knowing that $\psi$ is the Gaussian radial basis function and $\| \cdot \|_2$ is the Euclidean norm. These sources $\boldsymbol{\theta}_{\ell}, \ell = 1, \ldots, 9880$, differ from those used for constructing the surrogate models and are uniformly distributed over the entire domain. This spatial distribution was employed since we selected $p(\theta)$ to be the uniform distribution over $\mathcal{X}$ in score $\Gamma_{ED}$ (27).

The values $\Gamma_{ED}(\boldsymbol{\xi}_{10}^{\ell})$ are computed for all possible networks and the results are shown in Figure 6a. Allowing each network to include only one detector out of the three possible choices over each of the ten rectangular areas—see Figure 2—likely explains the periodic behavior. The optimization problem does not have a unique solution as seen from the expected values. Three different networks produce the largest score. Figure 6b shows the detectors locations of one of these three networks corresponding to the index 35,714.

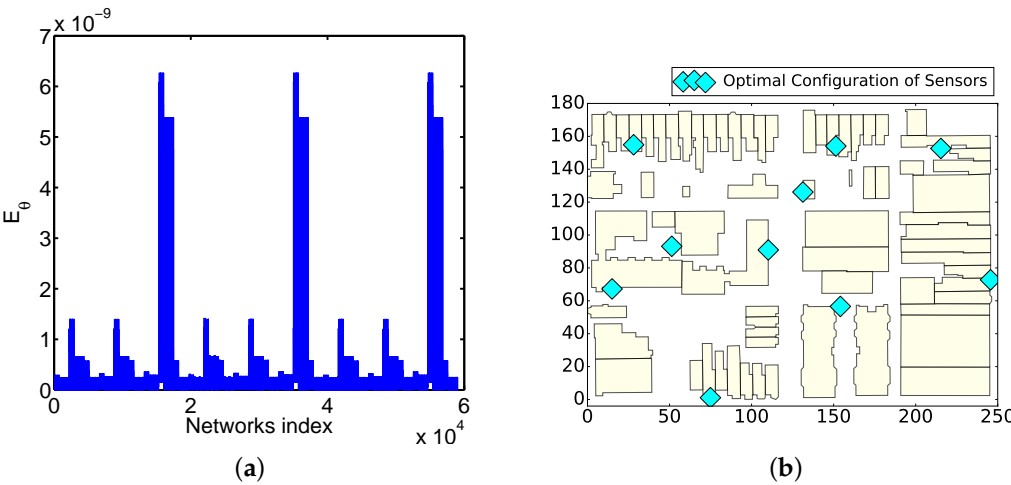

           (**a**)                                       (**b**)

**Figure 6.** (**a**) Scores $\Gamma_{ED}$ for all possible networks. The solution of the robust design problem is not unique; (**b**) The optimal network obtained as the solution of the robust design in the average sense strategy.

To test the obtained robust design network, we use the Formula (28) with the 11 randomly selected networks of sensors plotted in Figure 7. Next we set $\mathcal{M} = 50$, and uniform randomly select 50 true sources $\boldsymbol{\theta}_0^{\ell}$ from $\mathcal{X}$. Observations were then generated using the statistical model (9) for each source $\boldsymbol{\theta}_0^{\ell}$, $\ell = 1, \ldots, 50$ and network of sensors including the optimal one. We then solve the inverse problem (15) using the Particle Swarm algorithm [10] detailed in the Appendix A.

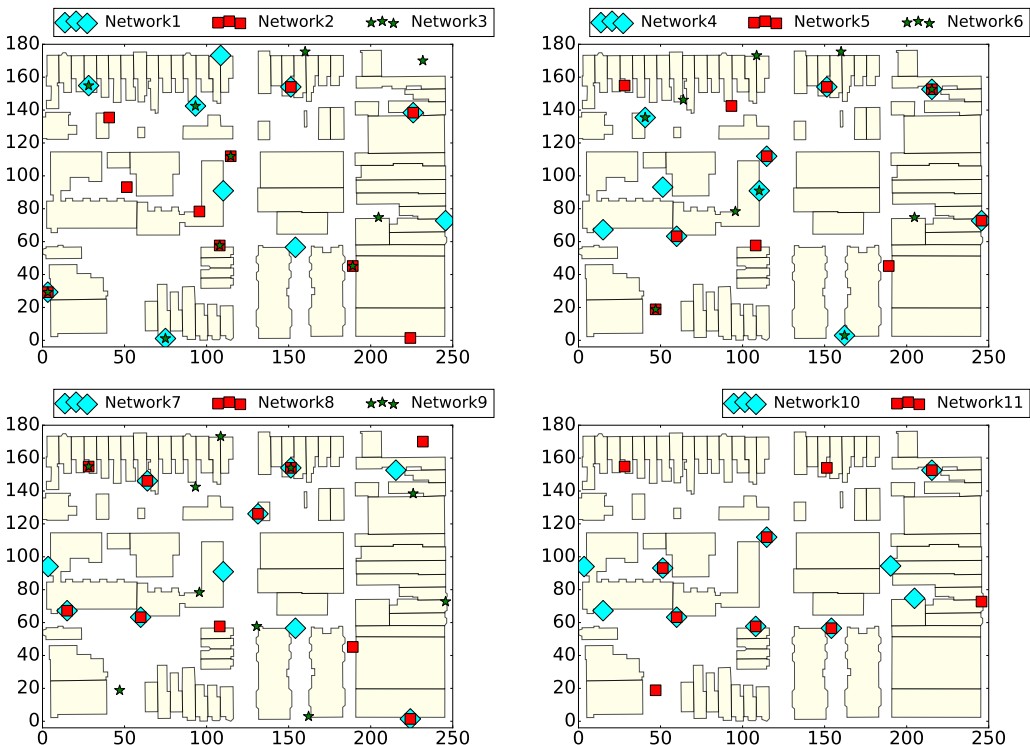

**Figure 7.** Random networks of sensors to be compared against the optimal network shown in the Figure 6.

The Particle Swarm approach is a global, meta-heuristic optimization algorithm motivated by social-psychological principles [27]. It was originally introduced in [10] and it was designed to imitate a social behavior such as the movements of birds in a flock or fishes in a shoal. Later the algorithm was simplified and its performance for solving optimization problems were reported in [28].

For our example, we set the inertia parameter to be 1.1 and the neighborhood of each particle is set to 4. The self and social adjustment coefficients $y_1$ and $y_2$ are set to 1.49. We select the swarm size to 70, and for each given source $\boldsymbol{\theta}_0^\ell$, $\ell = 1, \ldots, 50$, and network out of the 11 randomly selected networks plus the optimal one, we compute the inverse problem solution.

The errors of the inverse problem solutions (i.e., the estimated sources characteristics) are shown in Figure 8 for all possible networks and sources. We note that the solution obtained using the optimal network does not have the smallest error for all the sources. For example, for source number 28, the source characteristics errors obtained using the optimal network are larger than all the estimates errors associated with the random networks except Network 4. This is not unexpected, since the optimal design was obtained following an average sense formulation.

Next, the errors of the inverse problem solutions are averaged over the entire set of sources and the results of Formula (28) are illustrated in Figure 9 for all 12 considered networks. The index associated with the optimal network is 12 and corresponds to the smallest RMSE. This result suggests that we were able to identify the robust design in the average sense for the nuclear transport inverse problem.

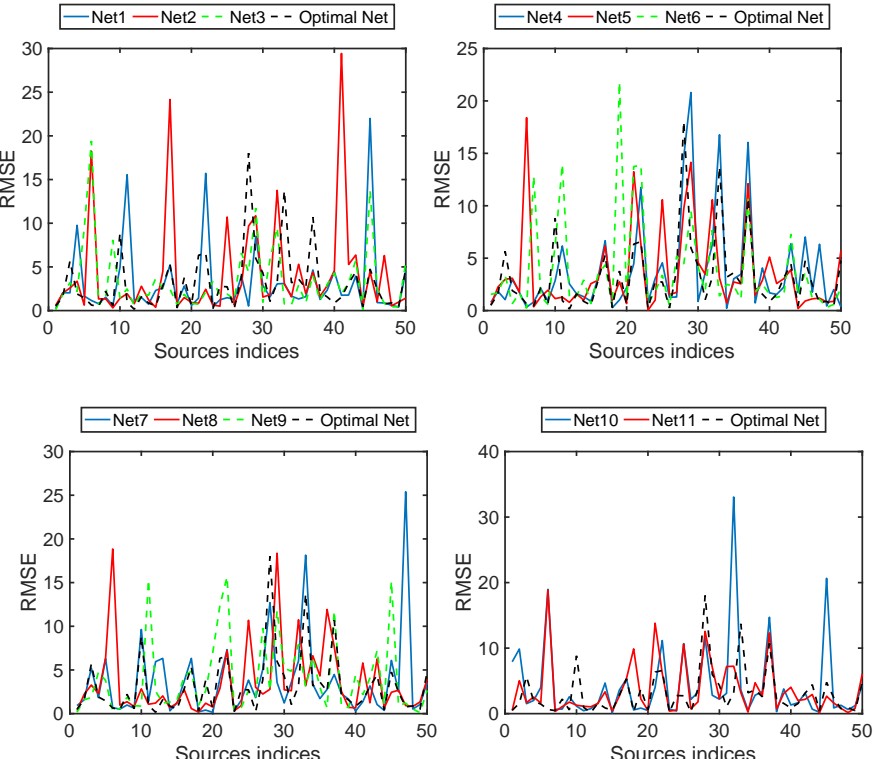

**Figure 8.** Root mean square errors for all the tested networks. The results are obtained for 50 sources whose components were selected from uniform distributions.

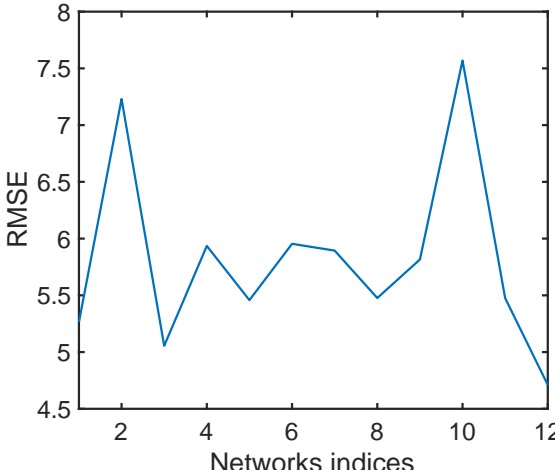

**Figure 9.** Root mean square errors averaged for all sources. The network corresponding to index 12 is the optimal one and has the smallest errors.

## 6. Conclusions and Future Work

In this investigation, we constructed a network of sensors to reduce uncertainty in the solution of a radiation detection inverse problem. We employed a robust design in the average sense method that eliminated the dependence on the true solution or *a priori* estimates. We focused on the ED-optimal design [15], whose solution maximizes the expected value of the determinant of the Fisher information matrix over the entire domain.

Since we generated a discrete number of possible sensors locations, the stochastic optimization problem is transformed into a combinatorial one. The number of possible networks decreased by imposing certain combination restrictions for the possible detectors' locations. We employed a radial

basis function surrogate model to alleviate the non-smoothness attributes of the radiation transport model due to the domain geometry.

By solving the combinatorial problem, we found that the solution of the optimization problem is not unique. However, we did identify multiple sensor networks that were optimal in a least- error sense. An optimal network associated with the largest ED-scores was compared against 11 randomly selected networks using a data set of 50 different sources. The overall RMSE revealed that the optimal network has more precision than any other network in the average sense.

In future work, we will reformulate the problem in a continuous framework; i.e., the number of detectors is unlimited inside the city environment. This formulation enables the use of very effective numerical algorithms. We will apply methods such as the stochastic Robbins—Monro algorithm [29,30].

**Author Contributions:** R.Ş. worked on conceptualization, methodology, software, validation and writing the manuscript. J.H. was responsible with some conceptualization details, software and reviewing the manuscript. J.C. helped with the results validation. R.C.S. and J.M. contributed to conceptualization, methodology and reviewing the manuscript.

**Funding:** This research was supported by the Department of Energy National Nuclear Security Administration NNSA Consortium for Nonproliferation Enabling Capabilities (CNEC) under the Award Number DE-NA0002576.

**Acknowledgments:** Razvan Ştefănescu thanks Michael Navon and Alen Alexanderian for their valuable suggestions regarding the current research topic.

**Conflicts of Interest:** The authors declare no conflict of interest. The funders had no role in the design of the study, the collection, analysis, and interpretation of data, the writing of the manuscript, and the decision to publish the results.

## Appendix A. Particle Swarm

The initialization stage of Particle Swarm is described in Algorithm A1. The algorithm starts by selecting the population size of the swarm denoted by $P$. Initially, the state positions $[\boldsymbol{\theta}^{old}]^j$ and velocities $[\boldsymbol{v}^{old}]^j$, $j = 1, 2, ..., P$ are randomly selected from uniform distributions; i.e., $[\theta_i^{old}]^j$, $[v_i^{old}]^j \sim U[l_i, u_i]$, $i = 1, 2, 3$. Each state point has an associated neighborhood of size $N = Ns$ influencing its future trajectory. Other parameters of the algorithm must be selected too, such as the inertia parameters $W^j \in \mathbb{R}$ and stall counter $c^j$ for $j = 1, 2, ..., P$. These parameters influence the space search.

---

**Algorithm A1** Particle Swarm—Initialization

---

1: Select swarm size $P \in \mathbb{N}$ and generate initial state points $[\boldsymbol{\theta}^{old}]^j$ and velocities $[\boldsymbol{v}^{old}]^j$, $j = 1, 2, ..., P$ such that $[\theta_i^{old}]^j$, $[v_i^{old}]^j \in [l_i, u_i]$, $i = 1, 2, 3$.

2: Select the minimum neighborhood size $minNs$ and the inertia parameters $W^j \in \mathbb{R}$, $j = 1, 2, ..., P$, $W^j \in [0.1, 1.1]$.

3: Set the stall counter $c^j = 0$ for all state points $j = 1, 2, ..., P$.

4: Set the self and social adjustment real variables $y_1$ and $y_2$.

5: Set $N = Ns$.

---

The evolution of the space point *jth* from the current state to the next one is described in Algorithm A2. The index notation is dropped. The proposal function depends on a two steps formula. First, the velocity $\mathbf{v}^{new}$ is adjusted via Equation (A1) while in the second phase, the new state is obtained by adding the newly generated velocity to its previous position (A2). The weights $y_1$ and $y_2$ denote the self and social adjustment coefficients steering the search towards either the state point $\mathbf{p}$ or its neighbors $\mathbf{g}$ best position. We denote the Hadamard product by $.*$.

A successful replacement of the best state point position $\mathbf{b}$ among the entire population ensures a change in the inertia parameter $W$ while a failure leads to a larger neighborhood selection and maintains $W$ constant. Finally, the new proposals are set to replace the current ones for the next iteration. The algorithm stops when the relative change in the lowest objective function value $J_b^*$ over a

range of predefined number of iterations is smaller than a specified tolerance, the maximum number of iterations is reached, or a preset objective function percentage decrease has been achieved.

---

**Algorithm A2** Particle Swarm—*jth* trajectory

---

1: Select $N$ state points other than $j$ to generate the associated neighborhood.

2: Set $flag = false$. Define set $S$ containing all the $N$ state points. Find the lowest objective function

$$\mathbf{g} = \min_{\boldsymbol{\theta}^{old} \in S} J(\boldsymbol{\theta}^{old}) \, and \, set \, J^{n*} = J(\mathbf{g}).$$

3: Select random vectors $\mathbf{u}_1$ and $\mathbf{u}_2$ of size 3 from the uniform distribution $U(0,1)$. Update the velocity:

$$\mathbf{v}^{new} = W \cdot \mathbf{v}^{old} + y_1 \cdot \mathbf{u}_1 . * (\mathbf{p} - \boldsymbol{\theta}^{old}) + y_2 \cdot \mathbf{u}_2 . * (\mathbf{g} - \boldsymbol{\theta}^{old}). \tag{A1}$$

4: Update the position

$$\boldsymbol{\theta}^{new} = \boldsymbol{\theta}^{old} + \mathbf{v}^{new}. \tag{A2}$$

5: Enforce the bounds. If any component of $\boldsymbol{\theta}^{new}$ is outside a bound, set it equal to that bound.

6: **if** $J(\boldsymbol{\theta}^{new}) < J^*$ **then** $\mathbf{p} = \boldsymbol{\theta}^{new}$, $J^* = J(\boldsymbol{\theta}^{new})$

7: **end if**.

8: **if** $J(\boldsymbol{\theta}^{new}) < J_b^*$ **then** $flag = true$, $J_b^* = J(\boldsymbol{\theta}^{new}) \, and \, \boldsymbol{b} = \boldsymbol{\theta}^{new}$, where $J_b^*$ corresponds to the smallest objective function in the swarm.

9: **else** $flag = false$

10: **end if**.

11: **if** flag = true **then** set $c = \max(0, c - 1)$ and $N = Ns$.

12:      **if** $c < 2$ **then** $W = 2 \cdot W$

13:      **end if**.

14:      **if** $c > 5$ **then** $W = W/2$ and ensure that $W$ is inside the bounds.

15:      **end if**.

16: **else** set $c = c + 1$, $N = \min(N + Ns, P)$

17: **end if**.

18: Set $\boldsymbol{\theta}^{old} = \boldsymbol{\theta}^{new}$ and $\boldsymbol{v}^{old} = \boldsymbol{v}^{new}$ and GO TO step 1.

---

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
