# Peer review of "Surrogate-Based Robust Design for a Non-Smooth Radiation Source Detection Problem"

_algorithms, doi:10.3390/a12060113_

Round 1
Reviewer 1 Report
In this article, the authors consider the optimal sensor placement problem to reduce the radiation sources detection error in a 2D environment with different coss-sections.
The paper is organized in different parts.
The authors detail the radiation source detection approach:
The observations are modeled as random variables.
The non-differentiability of the model is cured by approximating the detector responses model with a combination of radial basis functions whos coefficients are selected to ensure that the approximate model fit the observations. This model then provides an approximation of the detector response for a given radiation source (location+intensity).
For a given set of observations and considering the approximate response models, the radiation source is then determined using a maximum likelihood estimator.
The proposed optimal sensor placement is then explained:
The response model is linearized and surrogated in the min log-likelihood problem used for the source detection.
Considering the optimality condition of this problem, the optimal detectors' locations are determined.
As this first sensor placement is optimal for a given source location, the approach is then considered for all source locations in the domain, thus leading to a robust optimal sensor placement method.
Both the approximate response models, and the optimal sensor placement are evaluated in the experiment section.
The article is well written and sounds technically correct.
Author Response
Dear Reviewer,
Thank you for your comments concerning our manuscript entitled “Surrogate-Based Robust Design for a Non-Smooth Radiation Source Detection Problem”. We really appreciate taking your time to read our manuscript.
Razvan Stefanescu
Reviewer 2 Report
This work focuses on the study of advanced sensor positioning techniques for the detection of radiation in urban environments through sensor networks, using a simplified model of gamma radiation.
In the opinion of this reviewer, the contribution is relevant, interesting, and deals with topics little studied. However, although the paper seems technically correct, background and form corrections must be made before being accepted for publication:
The review of the state of the art should include a comparative analysis with other works, what are their weaknesses and the advantages of your work with respect to them. Please include references to other authors of similar works.
The terms "robust" and "uncertainty" are used in a non-usual sense throughout the document (search for example Optimization Under Uncertainty). Uncertainty is regularly used to describe changes in the parameters of the equation that represent some dynamics and not to measure the result of an optimization problem. It is recommended to use another term or define precisely what is meant by uncertainty.
On the other hand, the term "robust" is regularly used when there is uncertainty involved (robust control, robust optimization) but not in the sense used in this document; it is not demonstrated, or I can not understand, how this design can be called robust in the usual sense. My recommendation is that, since the contribution is sufficient, different terms should be used to avoid misleading the purpose of the paper, including the title (maybe optimal).
Although the authors correctly refer to [3] in obtaining the equation (1), it is necessary, for clarity of the document to define the operators apostrophe and arrow, or a more detailed description of the equation.
Line 60, the phrase: "The solution of (1) is typically computationally demanding and is not feasible for inverse analysis and uncertainty quantification methods requiring numerous model simulations" is ambiguous / unclear, it should be revised.
Line 71, the assumption "associated intensity by S0, which we assume is low enough that we can ignore time-dependent effects" is unclear, it should be detailed.
In line 72, the description of Omega is duplicated and delta should be described instead.
The parameter p for equation (10) is not described.
In equation (10), what is, and how L (calligraphic) is obtained? this should be included in the paper. Also, which method used to compute the coefficients lambda?
On line 121, a point is missing.
The use of Gaussian radial basis functions should be justified (line 122).
On line 124, there are undescribed notation and its calculating method.
Line 141 should be "...since one regularly does not..."
Line 153 and equation (28), if it is a quantitative measure then you should appoint and mention the correct name of such measure (please don't use goodness).
Line 166, the use of D-optimality criterion must be justified.
Lines 182-183, remove the apostrophes.
Line 202, the phrase "The number of possible networks is 30, 045, 015 as given by (30 choose 10)" is incomprehensible. The description/notation of networks is not explained.
Line 264, should be "...problem solutions (i.e. the estimated sources characteristics)..."
Line 279, regularly one does not use references in the conclusions.
Line 302, change to some like "The funders had no role in: the design of the study, the collection, analysis, and interpretation of data, the writing of the manuscript, neither in the decision to publish the results".
Author Response
Dear Reviewer,
Thank you for your letter and constructive comments concerning our manuscript entitled “Surrogate-Based Robust Design for a Non-Smooth Radiation Source Detection Problem”. We followed your comments carefully and made corrections which we hope to meet your expectation. The answers to the raised questions are provided in the attached document. We really appreciate taking your time to read our manuscript.
Razvan Stefanescu

Round 2
Reviewer 2 Report
The authors have addressed all my concerns and now provide a reference to the sense of robustness that their work presents. In the opinion of this reviewer the article is interesting and can be published with the changes made by the authors.